# Enhancement of prime editing via xrRNA motif-joined pegRNA

Guiquan Zhang[1,7], Yao Liu[2,7], Shisheng Huang[3,7], Shiyuan Qu[3], Daolin Cheng[1], Yuan Yao[4], Quanjiang Ji [5], Xiaolong Wang [2✉], Xingxu Huang [3,6✉] & Jianghuai Liu [1✉]

The prime editors (PEs) have shown great promise for precise genome modification. However, their suboptimal efficiencies present a significant technical challenge. Here, by appending a viral exoribonuclease-resistant RNA motif (xrRNA) to the 3′-extended portion of pegRNAs for their increased resistance against degradation, we develop an upgraded PE platform (xrPE) with substantially enhanced editing efficiencies in multiple cell lines. A *pan*-target average enhancement of up to 3.1-, 4.5- and 2.5-fold in given cell types is observed for base conversions, small deletions, and small insertions, respectively. Additionally, xrPE exhibits comparable edit:indel ratios and similarly minimal off-target editing as the canonical PE3. Of note, parallel comparison of xrPE to the most recently developed epegRNA-based PE system shows their largely equivalent editing performances. Our study establishes a highly adaptable platform of improved PE that shall have broad implications.

[1] State Key Laboratory of Pharmaceutical Biotechnology and MOE Key Laboratory of Model Animals for Disease Study, Model Animal Research Center at Medical School of Nanjing University, 210061 Nanjing, China. [2] Key Laboratory of Animal Genetics, Breeding and Reproduction of Shaanxi Province, College of Animal Science and Technology, Northwest A&F University, 712100 Yangling, Shaanxi, China. [3] Gene Editing Center, School of Life Science and Technology, ShanghaiTech University, 100 Haike Rd., Pudong New Area, 201210 Shanghai, China. [4] Hangzhou Global Scientific and Technological Innovation Center, Zhejiang University, 311215 Hangzhou, China. [5] School of Physical Science and Technology, ShanghaiTech University, 100 Haike Rd., Pudong New Area, 201210 Shanghai, China. [6] Zhejiang Laboratory, 311100 Hangzhou, China. [7] These authors contributed equally: Guiquan Zhang, Yao Liu, Shisheng Huang. ✉email: xiaolongwang@nwafu.edu.cn; huangxx@shanghaitech.edu.cn; liujianghuai@nju.edu.cn

Harnessing the CRISPR-Cas system to empower precise genome modification hold great promise to revolutionize medicine and agriculture. The recently emerged CRISPR-based prime editors (PEs) represent a major technological breakthrough, enabling installation of various point mutations and small insertions/deletions, while circumventing the requirement of double-stranded DNA breaks (DSB)[1]. The basic PE system consists of a fusion protein of Cas9 (H840A) nickase (nCas9) and a reverse transcriptase (RTase) domain, together with an engineered prime editing guide RNA (pegRNA). The pegRNA differs from a common sgRNA by an extended 3′ region containing a primer-binding site (PBS), in conjunction with an adjacent sequence of reverse transcription template (RT template). Here, the PBS is poised to hybridize with the bases upstream of the nCas9 (H840A)-generated nick, while the RT template encodes the genetic information of the intended edits and directs reverse transcription. To manipulate the ensuing cellular DNA repair pathway for productive incorporation of intended edits, a second single-guide RNA (sgRNA) is used for nicking the strand opposite to the RT action[1]. In aggregate, this forms a readily applicable platform named PE3 (in relation to a more basic version of PE2 without the use of a second sgRNA).

Since its introduction, PE has been further applied to genome modification in rice, wheat, zebrafish and mouse embryos[2–5]. Notably, the efficiencies of current PE are generally less satisfactory, which has stimulated independent efforts for its improvement. For instance, a recent report adopted a dual-pegRNA strategy which led to notably higher PE efficiencies in rice[6]. Another study reported an enhancement of prime editing efficiency in cell lines and mice by the inclusion of a proximal dead sgRNA and by fusion of chromatin-modulating peptides to the nCas9-RTase moiety[7]. Nevertheless, it is worth noting that even with these different innovative designs, PE efficiencies at most tested genomic loci remained suboptimal, indicating existence of other technical "bottlenecks". We have also recently developed an alternative approach of enhanced prime editing system (ePE), taking advantage of a version of a pegRNA that is processed and bound by Csy4 at the 3′ end[8], indicating another independent avenue for improving PE efficiencies by modifying the overall structure of pegRNAs.

As the 3′ extended PBS and RT template region in the pegRNA is not bound by the nCas9 structure, it may be susceptible to degradation, or to formation of unproductive secondary structures. It is conceivable that Csy4 processing/binding of pegRNA in the ePE system may contribute to alleviating such setbacks[8]. However, introduction of a Csy4 protein could lead to further delivery and cytotoxicity issues[8,9]. Therefore, we considered incorporation of other stabilizing RNA motifs to pegRNA for its activity enhancement. The Xrn1-resistant RNAs (xrRNAs) are a group of conserved structures found in flaviviruses, including Dengue, Yellow fever, West Nile, and Zika (Fig. 1a and Supplementary Fig. 1)[10–12]. Located at the beginning of the 3′ untranslated region (3′-UTR) of the viral genome, such structure protects the downstream viral RNA from degradation by the 5′→3′ exoribonuclease Xrn1, resulting in the production of a non-coding sub-genomic viral RNA that functions to enhance viral pathogenicity[10]. The xrRNAs adopt a characteristic knot-like structure that is thought to mechanically impede Xrn1 processing from the 5′ direction[11,13,14]. Importantly, recent evidence demonstrated that even under bidirectional pulling forces, the xrRNA motif exhibited a remarkably high level of mechanical rigidity and resistance to unfolding[15]. Therefore, we envisioned that appending an xrRNA motif to the 3′ end of pegRNA may promote its stability/activity. Here we developed an xrRNA-joined pegRNA prime editor (xrPE), which showed markedly enhanced prime-editing efficiencies. Our study presents an improved PE system that is readily applicable, and has implications for future advancements of precise genome editing.

## Results

### The xrRNA-joined pegRNAs show enhanced prime editing activities toward a reporter.
Given the structural and mechanical features of xrRNAs[10,11,15], adding a xrRNA motif to the 3′ end of a pegRNA appear a reasonable strategy to improve the latter's stability/activity. Here, five xrRNA motifs from different flaviviruses (Murray Valley encephalitis (MVE), West Nile virus (WNV), Zika, Dengue (Dengue), and Yellow Fever (YF)) were selected for testing (Fig. 1a and Supplementary Fig. 1).

For convenience, we first constructed a plasmid-borne editing reporter in the format of mRuby-linker-EGFP (Fig. 1b and Supplementary Fig. 2). The linker contains a "TAG" stop codon to prevent EGFP coding region from translation. A prime editing-programmed "TAG" to "TGG" conversion (A-to-G base transition) would lead to further translation of EGFP (as a mRuby fusion partner), whose fluorescence would indicate successful prime editing events. Therefore, we next constructed wild-type (WT) pegRNA and five different viral xrRNA-joined pegRNAs (xr-pegRNAs) targeting the reporter above (Supplementary Figs. 1 and 3a).

The initial investigations were made using a single nicking-dependent PE2 system. Human embryonic kidney (HEK) 293 T cells were co-transfected with plasmids encoding PE2, different pegRNAs, and the editing reporter. 48 h after transfection, we observed varied levels of EGFP[+], prime-edited cells in our experimental groups by fluorescence microscopy as well as by flow cytometry (Fig. 1c and Supplementary Fig. 3b). Subsequent quantifications of flow cytometry revealed that on average, these xr-pegRNAs triggered 37.2% (MVE), 29.7% (WNV), 33.4% (Zika), 23.3% (Dengue), and 25.3% (YF) of EGFP positivity (relative to mRuby), with 4 out of 5 groups (except the Dengue group) showing statistically significant enhancement over the control rate of 20.0% induced by WT pegRNA (Fig. 1d). Such patterns of EGFP fluorescence roughly correlated with those determined by Western blot (Fig. 1e). To more accurately determine the rate of prime editing in these groups, DNA samples were extracted from the further sorted mRuby[+] cells, amplified by polymerase chain reaction (PCR) around the "TAG" target and analyzed by next-generation sequencing (NGS). The results showed that different xr-pegRNAs (in the same order as above) increased the efficiencies of prime editing at this particular target to levels 2.8-, 2.6-, 2.4-, 1.2-, or 1.8-fold of that induced by the WT pegRNA, with 4 out of 5 testing groups exhibiting statistically significant enhancements (Fig. 1f and Supplementary Fig. 3c). The levels of edit:indel ratios, which indicate the degrees of editing accuracy, showed generally correlative increases in the xr-pegRNA groups at this plasmid-borne target site (Supplementary Fig. 3d). Sanger sequencing of the amplicons further confirmed the activity patterns by different constructs of pegRNAs (Supplementary Fig. 3e). Collectively, these results demonstrated that pegRNAs adjoined by different viral xrRNA motifs generally enabled higher prime editing efficiency in a reporter, although to some varied degrees.

### The xr-pegRNA enhances prime editing of base conversions at various sites within genomic context.
Next, we asked whether the xr-pegRNA could improve prime editing at sites within the genomic context. WT pegRNAs and xr-pegRNAs were designed to target 6 human gene loci (i.e., ALDOB, RIT1, EMX1, FANCF, RNF2, or HEXA) for various base conversions. For prime editing (PE2), we transfected HEK293T cells with the pCMV-PE2 plasmid and the plasmids encoding the pegRNAs. The transfected

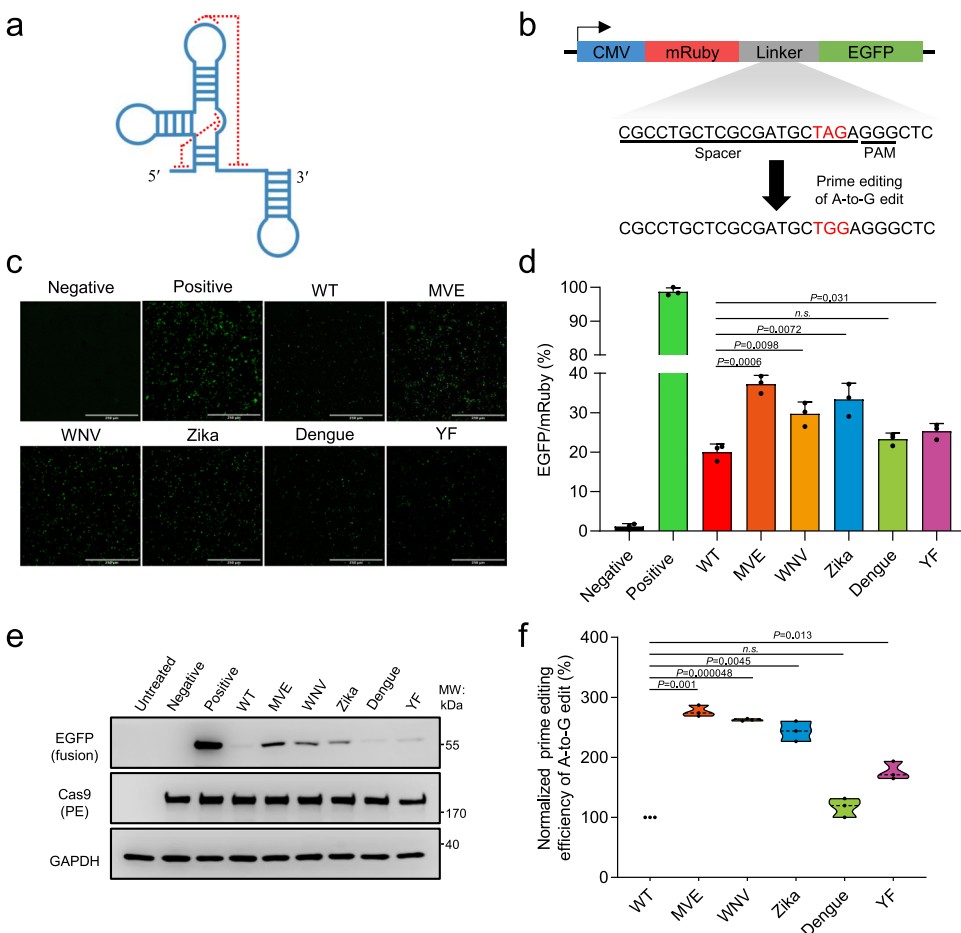

**Fig. 1 The xrRNA-joined pegRNAs show enhanced prime editing activities toward a reporter. a** The schematic secondary structure of a representative xrRNA motif. Some long-distance interactions (highlighted by red dotted lines) contribute to the formation of a stable knot-like structure. **b** Illustration for the fluorescent prime editing reporter system. The translation of EGFP sequence as part of a mRuby-led fusion protein is prevented by a stop codon (TAG, red). Prime editing-mediated of A-to-G edit would allow the expression of mRuby-EGFP fusion protein. The spacer sequence and the PAM for prime editing are underlined. **c**. The xrRNA motifs from five different viruses: Murray Valley encephalitis (MVE), West Nile virus (WNV), Zika, Dengue (Dengue), and Yellow Fever (YF)) were appended to the 3′ end of pegRNAs that targets the reporter. In results shown in **c–f**, HEK293T cells were co-transfected with plasmids for PE2, WT or modified pegRNA, and the reporter. EGFP⁺ cells were observed under a fluorescent microscope. Transfection of PE2, a non-targeting pegRNA and the reporter served as the negative control, whereas in the positive control the reporter plasmid was replaced with one encoding a constantly expressing mRuby-EGFP fusion protein. Scale bars, 250 μm. **d** Following prime editing using WT pegRNA and xr-pegRNAs, the frequencies (%) of EGFP⁺ (relative to mRuby⁺) were measured by flow cytometry. **e** Following prime editing using WT pegRNA and xr-pegRNAs, the expression of EGFP was determined by Western blot. **f** The reporter-targeted editing efficiencies were determined by deep-sequencing of DNA prepared from mRuby⁺ cells. The editing frequencies induced by PE with WT pegRNA were set as 100%. In quantitation shown in **d** and **f**, data are presented as mean values ±SD, n = 3 biological replicates. Two-tailed Student's t tests (one-sample test for **f**) were performed (P values are marked on the graphs, n.s. not significant). The P values [n.s.] not marked on **d** and **f** are 0.09 and 0.20, respectively. Source data are provided as a Source Data file.

cells (72 h) were sorted based on a EGFP marker within the pegRNA constructs (Supplementary Fig. 4a), and the genomic DNA samples were prepared. The amplified PCR products were analyzed using NGS. The 3′-joining by different xrRNA motifs generally improved the efficiencies of PE at all 6 gene loci (Fig. 2a and Supplementary Fig. 4b). When different xrRNA modifications were considered as one experimental group, they showed higher levels of activities over the parallel WT pegRNA group, averaged between 1.9-fold [EMX1] and 1.1-fold [RNF2] for a given target site. Importantly, the levels of corresponding edit:indel ratios at these sites were either minimally changed or improved (Supplementary Fig. 6a). Additionally, when the efficiencies at all these 6 gene loci were considered as a whole [expressed as "median (mean)" throughout this report], modifications of pegRNAs by different viral xrRNAs could each increase the overall efficiencies of PE2-mediated base conversion to levels between 1.5 (1.5)- and 1.2 (1.2)-fold of that by WT

pegRNA (Fig. 2b). Here, the 3′ Zika xrRNA motif appeared to confer the best overall enhancement effect.

We further investigated the performance of xr-pegRNAs in the context of PE3 that features a second nicking of the unedited stand (with an additional nick-sgRNA) to increase productive incorporation of templated edits[1]. The same 6 loci as above were tested and analyzed by targeted NGS. The xrRNA-joining of pegRNAs significantly increased the efficiency of PE3 at various loci (Fig. 2c and Supplementary Fig. 5), to average levels ranging between 3.5-fold [EMX1] and 1.2-fold [RNF2] of those by WT pegRNAs (different xrRNA modifications considered as one group). Here, compared to the control pegRNA group, the use of xr-pegRNAs were generally associated with minimally changed or improved edit:indel ratios at a given site (Supplementary Fig. 6b). It is worth mentioning that at certain sites (e.g., at EMX1 and HEXA), PE3 efficiencies under the context of the WT pegRNA did not appear superior to PE2 (Fig. 2a, c), partially attributed to

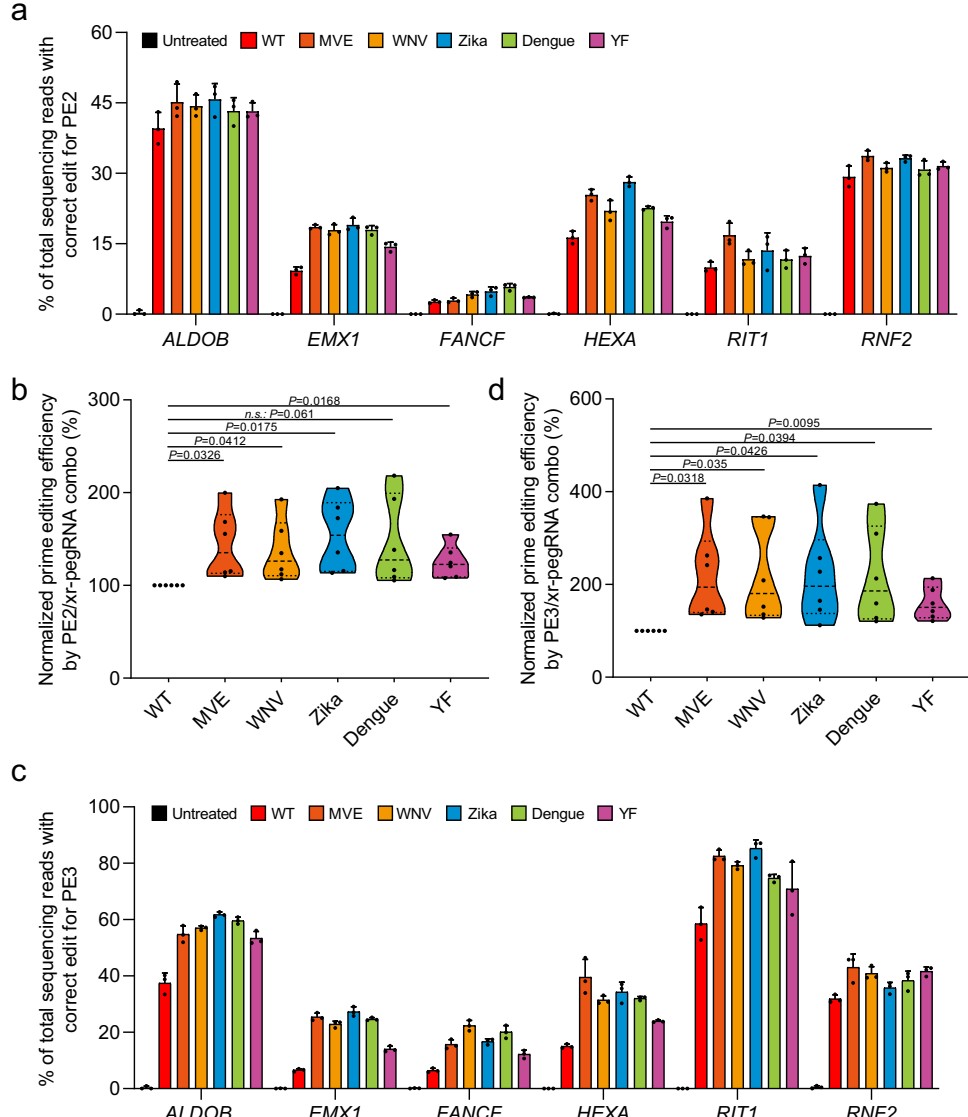

**Fig. 2 The xr-pegRNA enhances prime editing of base conversions at various sites within genomic context. a** HEK293T cells were co-transfected with plasmids for PE2 and WT pegRNAs or different xr-pegRNAs targeting indicated (6) genomic loci for base conversions. Following isolation of genomic DNA from transfected cells (sorted), correct editing rates at each site were determined by deep-sequencing (mean ± SD, $n = 3$ biological replicates). Reads that only contain the intended edits were counted. **b**. Results in **a** is further analyzed by considering editing at all sites ($n = 6$ sites) as a whole. The editing frequencies induced by PE2 with WT pegRNA were set as 100%. **c**. Experiments were carried out similar to (**a**), except that a PE3 strategy was used (mean ± SD, $n = 3$ biological replicates). **d** Results in **c** is further analyzed by considering editing at all sites ($n = 6$ sites) as a whole. The editing frequencies induced by PE3 with WT pegRNA were set as 100%. In the violin plots shown in **b** and **d**, each point represents the averaged editing activity at the particular site. The thicker dotted line shows the medians of all data points, while the thinner dotted lines correspond to quartiles (1st and 3rd). Two-tailed one-sample Student's *t* tests were performed. The *P* values are marked on the graphs (n.s. not significant). Source data are provided as a Source Data file.

the higher incidence of indel introduction by PE3 (see Supplementary Fig. 6b legend)[1]. Indeed, with the bioinformatics pipeline used throughout this study, reads harboring simultaneous correct editing and other undesired insertion/deletion were considered only as indels. We noted that at such sites, the use of xr-pegRNAs subsequently enabled PE3 to outperform PE2 (Fig. 2a, c).

When prime editing at all six gene loci were considered as a whole, pegRNA 3′-modifications by these five different viral xrRNAs could each enhance PE3 efficiencies to levels between 2.0 (2.2)- and 1.5 (1.6)-fold of that by WT pegRNA (Fig. 2d). Notably, the use of xr-pegRNA with PE3 generally led to a higher degree of activity enhancement than with PE2 (Fig. 2b, d). Consistent with the patterns of the PE2 results (Fig. 2b), here the

3′ Zika xrRNA motif also appeared to confer the best overall activity improvements, while the enhancement effects by xrRNA elements from the MVE and Dengue were just closely behind (Fig. 2d). Moreover, a parallel assessment of edit:indel ratios by each group of xr-pegRNAs relative to the control group showed either minimally changed or improved levels, with unapparent differences between different xrRNA groups (Supplementary Fig. 6c). Collectively, these experiments have indicated the potential of enhancing prime editing by the use of xr-pegRNA. Our subsequent experiments would focus on the use of one xrRNA motif (the one derived from Zika virus) for further pegRNA modification, as it had consistently provided the best overall enhancement effects on PE2- and PE3-mediated editing of genomic loci (Fig. 2b, d).

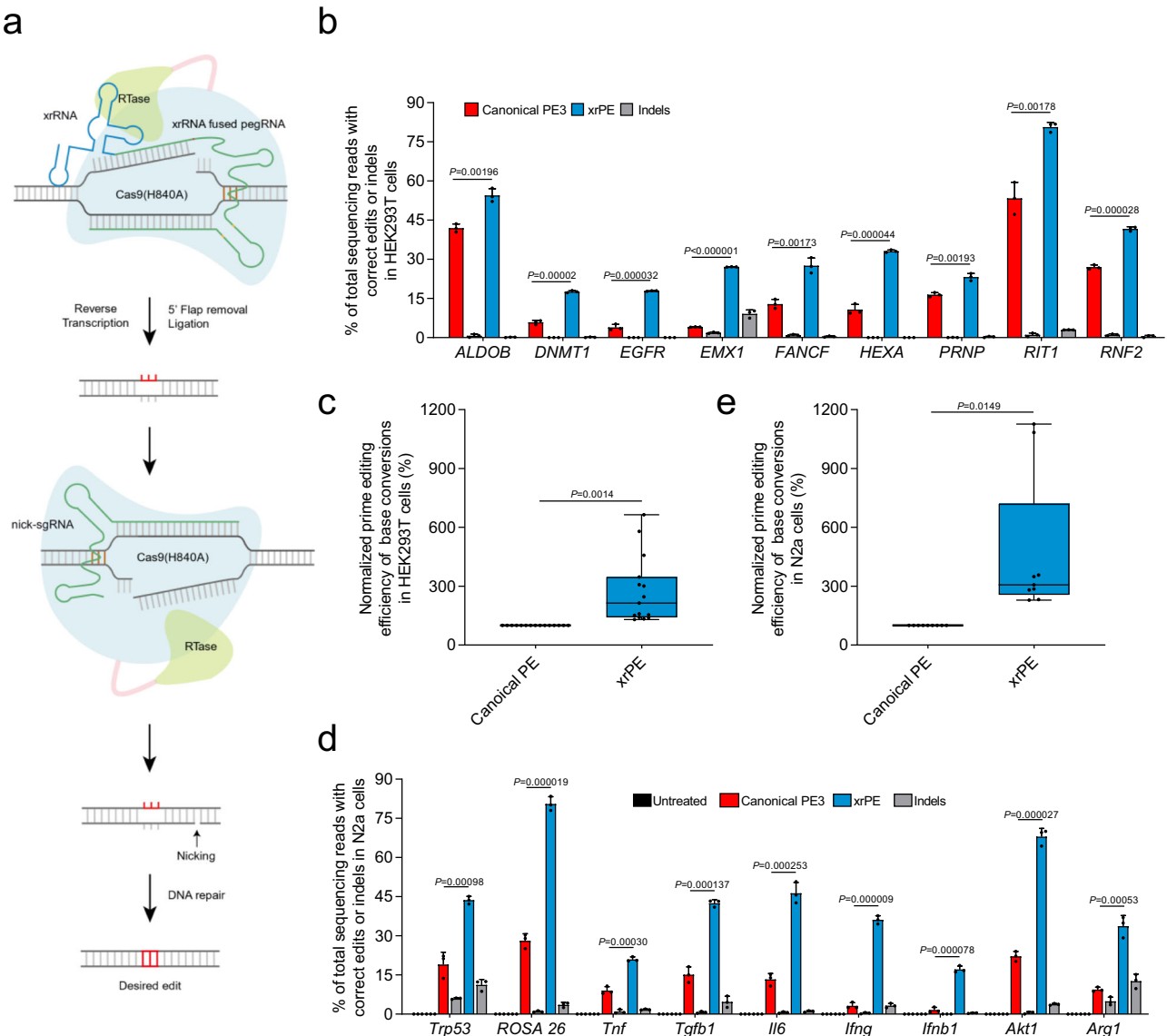

**Fig. 3 The xrPE shows enhanced performance for base conversions in multiple cell types. a** An illustration for the xrPE platform. The joining of an xrRNA motif (Zika) to the 3' end of pegRNA is shown. A fusion protein of Cas9 H840A nickase and a reverse transcriptase (Moloney Murine Leukemia Virus, M-MLV) is guided by the modified pegRNA to a DNA target. The yellow marks within xrRNA-joined pegRNA indicate an alternative C-G base pair replacing a U-A in the main scaffold, to potentially reduce premature termination[8]. The prime editor nicks the DNA and reverse transcribes using the 3'-extended portion of pegRNA as the template. This is followed by 5' flap removal and ligation to complete editing on one strand. When supplying another sgRNA to nick the non-edited strand, the cellular DNA repair mechanisms tend to install the desired edit into the genome. **b** HEK293T cells were transfected with plasmids for canonical PE3 or xrPE for base conversion at 9 individual sites as indicated. Correct editing efficiencies were determined by deep-sequencing (mean ± SD, $n = 3$ biological replicates). For targets same as those in Fig. 2c, a consistent pattern of activity enhancements is noted. Gray bars next to those for PE3 (red) and xrPE (blue) indicate the indel frequencies associated with each tool. **c** Results in **b** and Supplementary Fig. 8a are further analyzed by considering editing at all sites ($n = 15$ sites) as a whole. The editing frequencies induced by canonical PE3 were set as 100%. **d** The experiment similar to **b** was carried out in N2a cells (base conversions at 9 individual sites). The rates for correct editing and indel formation are shown (mean ± SD, $n = 3$ biological replicates). **e** Results in **d** are further analyzed by considering editing at all sites ($n = 9$ sites) as a whole. The editing frequencies induced by canonical PE3 were set as 100%. Multiple t tests (two-tailed) were performed in data from **b**, **d**. Discoveries were determined using the two-stage linear step-up procedure of Benjamini, Krieger, and Yekutieli, with $Q = 1\%$. When discoveries are made (9/9 in **b** and 9/9 in **d**), the exact $P$ values (unadjusted) are shown on the graphs. In the box plots shown in **c**, **e**, each data point represents the averaged editing activity at the particular site. The center line shows medians of all data points and the box limits correspond to the upper the lower quartiles, while the whiskers extend to the largest and smallest values. Two-tailed one-sample Student's t tests were performed (with $P$ values marked). Source data are provided as a Source Data file.

**The xrPE shows enhanced performance for various mutation types in multiple cell types**. After the initial developments, we next focused on prime editing with the Zika xrRNA-joined pegRNA under a PE3 framework (Fig. 3a), which we termed xrPE. We further benchmarked the performance of xrPE for base conversion in HEK293T cells targeting a larger panel of 15

genomic sites. Editing efficiencies were determined as described above. Compared to canonical PE3, xrPE led to statistically significant improvements in PE efficiencies for 13 out of 15 sites (except for the sites in *CCR5* and *PD1*) (Fig. 3b and Supplementary Figs. 7 and 8a). When all 15 sites were considered as a whole, the levels of xrPE-driven base conversion were on average

2.1 (2.8)-fold higher than those by canonical PE3 [ranging between 6.6- and 1.3-fold] (Fig. 3c). At these sites, a potential trend of greater edit:indel ratios in the xrPE groups (vs. PE3) were observed (Supplementary Fig. 8b, c, median fold-change: 1.9), while the data featured apparent variability. To further substantiate the observed enhancement effect, we next compared the performance of xrPE and canonical PE3 with different lengths of RT templates as variables, as RT template length represents one variable impacting prime editing efficiencies[1,8]. Notably, xrPE outperformed canonical PE3 to largely similar degrees, independent of the RT template length (Supplementary Fig. 9).

Another human cell line, i.e., HeLa, was subjected to prime editing of base conversion at eight different target loci using canonical PE3 and xrPE. The xrPE increased the prime editing efficiencies to an overall level of 2.7 (3.3)-fold of that by canonical PE3 [between 5.8- and 1.4-fold, with 5 out of 8 comparisons of statistical significance] (Supplementary Fig. 10a–c). A parallel assessment of edit:indel ratios by xrPE at these sites in comparison to PE3 showed an overall equivalent level (Supplementary Fig. 10d, median fold-change: 1.0). In addition, in mouse-derived N2a cell line, we found that xrPE outperformed canonical PE3 for base conversion in 9 out of 9 tested sites. The overall efficiencies of prime editing by xrPE at these 9 sites were 3.1 (4.7)-fold higher than those by canonical PE3 [ranging between 11.3 and 2.3-fold] (Fig. 3d, e and Supplementary Fig. 11), with generally no apparent changes in edit:indel ratios (Supplementary Fig. 8d, median fold-change: 1.3).

PE presents a clear advantage over other genome editing tools in its ability to install precise small insertions and deletions into the genome without the requirement of DSB, which has broad implications including the potential for treatment of related genetic diseases[1]. Therefore, we next extended xrPE to the application of introducing small insertions/deletions. We designed pegRNAs and xr-pegRNAs for either 3-bp deletions or insertions at 6 different human gene loci. In HEK293T cells, compared to canonical PE3, xrPE showed higher efficiencies in 4 out of 6 sites for deletions, and 6 out of 6 sites for insertions. When all sites were considered, the overall efficiencies by xrPE for deletion and insertion in HEK293T cells were 1.8 (1.9)-fold [up to 3.8] and 2.5 (2.6)-fold [up to 5.5] higher than those by canonical PE3, respectively (Fig. 4a, b). The overall edit:indel ratios associated with these xrPE-mediated modifications were equivalent to the PE3 groups (Supplementary Fig. 12a, median fold-changes: 1.0 and 0.9 for small deletions and insertions, respectively).

For the same 6 sites in HeLa cells, xrPE also exhibited higher efficiencies than canonical PE3 for both deletions (6 out of 6) and insertions (4 out of 6). When results from all 6 sites in HeLa cells were considered, average levels of enhancement were 3 (3.3)-fold [up to 5.9] for deletions and 2.5 (2.6)-fold [up to 4.1] for insertions (Supplementary Fig. 13a, b), with overall no apparent changes in edit:indel ratios (Supplementary Fig. 13c, median fold-changes: 1.3 and 0.9 for small deletions and insertions, respectively). In addition, 6 individual sites in N2a cells were selected for prime editing-driven deletions and insertions. Notably, at all 6 sites and for both mutation types, xrPE showed higher efficiencies than canonical PE3. An overall pattern of 4.5 (4.6)-fold [up to 9.1], and 2.2 (2.1)-fold [up to 2.5] increases was respectively observed for deletions and insertions (Fig. 4c, d). The parallel edit:indel ratios were not apparently affected (Supplementary Fig. 12b, median fold-changes: 1.0 and 0.7 for small deletions and insertions, respectively). Taken together, xrPE shows significant improvements in efficiency over canonical PE3 for installing various types of genetic modifications in multiple cell types.

**Editing fidelity by xrPE remain comparable to canonical PE3.** The editing fidelity is an essential parameter for genome editing tools. On-target byproducts represent a major category of imprecise editing by PE[1]. Given the notably enhanced editing efficiencies by xrPE in comparison to PE3, we next directly considered the corresponding on-target indel rates. For instance, the indel percentages in association with targeted base conversions, small deletions, or small insertions by either PE3 or xrPE in HEK293T cells were directly compared (related to Supplementary Fig. 8a and Figs. 3b and 4a). Overall fold-changes (medians) of 2.4, 1.8, and 2.5 were seen in indel rates associated with base conversions, small deletions, and small insertions, respectively (Supplementary Fig. 14a–c). Although some of the differences from these comparisons did not reach statistical significance ($p = 0.15$, $p = 0.13$ and $p < 0.05$, respectively), such patterns of indel rates by xrPE in reference to the PE3 groups roughly correlate with the xrPE-associated 2.1-, 1.8-, and 2.5-fold of overall efficiency enhancements for the 3 types of modifications (see Figs. 3c and 4b). This is consistent with the generally unapparent changes in edit:indel ratios at sites subjected to modifications by either xrPE or PE3 in all cell types tested (see Supplementary Figs. 8b–d, 10d, 12a, b, and 13c).

Besides indels, undesired by-products by prime editing may potentially also include base conversions. Additional analyses were carried out to determine the base frequency within 10-bp of the nCas9 cleaving sites guided by the spacer sequences (e.g., in results related to Fig. 3b and Supplementary Fig. 8a). When 15 edited sites in HEK293T cells were analyzed, we found no unintended base conversion byproducts by either canonical PE3 or xrPE group (Supplementary Fig. 15).

PE is associated with relatively low off-target activities, conceivably attributed to the restrictions by multiple events of strand hybridizations[1]. Other studies validated the low off-target activities by PE[16,17]. It would be important to establish whether xrPE preserves the good targeting specificity of PE. Three editing applications were carried out in HEK293T cells with canonical PE3 and xrPE (targeting *EMX1*, *FANCF,* and *HEXA*). Off-target sites for each pegRNA and nick-sgRNA were predicted by Cas-OFFinder[18], followed by selection of 8 higher probability sites per individual guide sequence. No editing-associated indel formations were detected at these sites in either the canonical PE3 or xrPE groups (Supplementary Fig. 16), indicating that xrPE inherits the good target specificity of the PE platform.

**The xr-pegRNAs introduced via a lentiviral vector also showed improvements in activities over the original pegRNA.** Lentiviral vector (LV)-mediated introduction of prime editing reagents widens their applications. Therefore, we further explored whether the xr-pegRNA could show enhanced activities upon transduction by an LV. To this end, we packaged the WT pegRNAs and xr-pegRNAs against the *FANCF* (+1 ACT insertion), *HEXA* (+1A-to-G), *RIT1* (+5G-to-A), *PRNP* (+1–3 deletion), or *RNF2* (+1–3 deletion) sites into LVs and subsequently transduced the HEK293T cells. The transduced cells were subsequently transfected with a PE2 plasmid containing a GFP label. The editing efficiencies were later determined in GFP+ cells sorted by flow cytometry. At each of these sites (except for *RNF2*), the LV-introduced xr-pegRNAs resulted in strong improvements of editing efficiencies compared to the WT pegRNAs. Across all these tested sites, an overall 5.4 (4.8)-fold efficiency improvement was observed (Supplementary Fig. 17a, b). Interestingly, the edit:indel ratios associated with the LV-delivered xr-pegRNAs were apparently higher compared to those from the pegRNA groups (Supplementary Fig. 17c, at all but the *RNF2* site), which

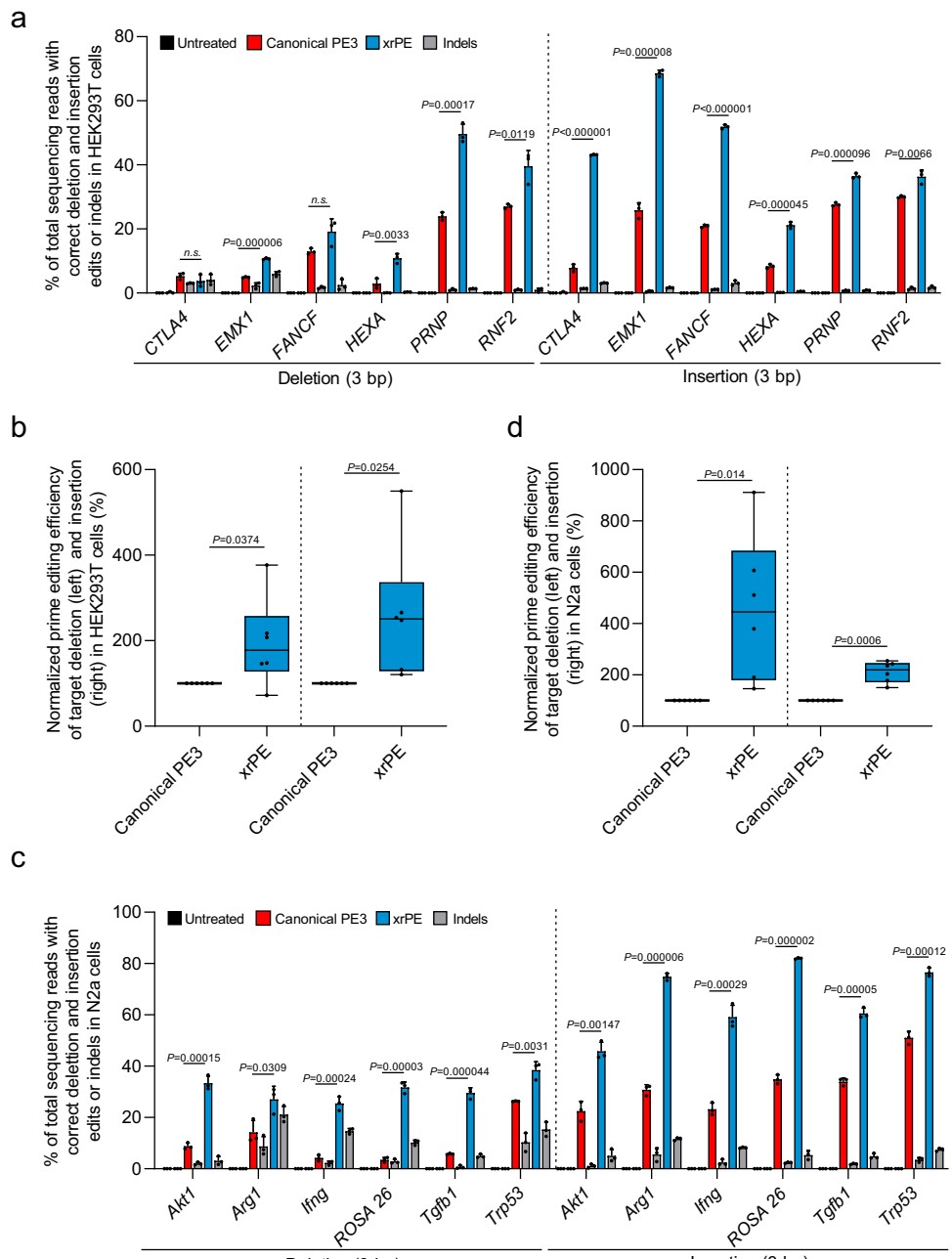

**Fig. 4 The xrPE shows enhanced performance for precisely introducing small deletions and small insertions in multiple cell types. a** HEK293T cells were transfected with plasmids for canonical PE3 or xrPE targeting 6 individual sites for 3-bp deletions and 3-bp insertions, separately. Correct editing efficiencies were determined by deep-sequencing (mean ± SD, $n = 3$ biological replicates). Gray bars next to those for PE3 (red) and xrPE (blue) indicate the indel frequencies associated with each tool. The same set of indel data from untreated cells (background) were presented for deletions and insertions. **b** Results in **a** is further analyzed by considering editing at all sites ($n = 6$ sites for deletions and insertions, respectively) as a whole. The editing frequencies induced by canonical PE3 were set as 100%. **c** The experiment similar to **a** was carried out in N2a cells (at 6 individual sites for 3-bp deletions and 3-bp insertions, respectively). The rates for correct editing and indel formation are shown (mean ± SD, $n = 3$ biological replicates). The same set of indel data from untreated cells (background) were presented for deletions and insertions. **d** Results in **c** were further analyzed by considering editing at all sites ($n = 6$ sites for deletions and insertions, respectively) as a whole. The editing frequencies induced by canonical PE3 were set as 100%. Multiple $t$ tests (two-tailed) were performed in data from **a**, **c**. Discoveries were determined using the two-stage linear step-up procedure of Benjamini, Krieger, and Yekutieli, with $Q = 1$%. When discoveries are made (10/12 in **a** and 12/12 in **c**), the exact $P$ values (unadjusted) are shown on the graphs. Otherwise, the comparisons are marked by n.s. not significant, where the corresponding $P$ values are 0.315 ($-3$ bp in *CTLA4*) and 0.062 ($-3$ bp in *FANCF*), respectively. In the box plots shown in **b**, **d**, each data point represents the averaged editing activity at the particular site. The center line shows medians of all data points and the box limits correspond to the upper the lower quartiles, while the whiskers extend to the largest and smallest values. Two-tailed one-sample Student's $t$ tests were performed (with $P$ values marked). Source data are provided as a Source Data file.

suggest a selective enhancement of PE2-mediated precise genome editing.

Additionally, coupled with a second-nicking mechanism (PE3 strategy), the LV-introduced xr-pegRNAs also robustly increased the editing efficiencies at the same 4 out of 5 tested loci (except for *RNF2*). Herein, an overall 4.2 (4.3)-fold efficiency enhancement by xr-pegRNA over pegRNA was observed (Supplementary Fig. 18a, b), with a potential trend of higher edit:indel ratios associated with the use of xr-pegRNA (Supplementary Fig. 18c). The results confirm that xr-pegRNAs introduced via LV also out-perform pegRNA to drive more efficient prime editing.

**Mechanistic examination of pegRNA activity enhancement by 3′-joining of xrRNA motif.** The clearly improved activities of xr-pegRNAs may be attributed to xrRNA structure-dependent 3′-protection of the pegRNAs from degradation by exonucleases. Indeed, just prior to the initial submission of our study, Nelson JW et al. reported a strategy of using engineered pegRNA (termed as "epegRNA") to improve PE[19]. In particular, the work mainly focused on the 3′-joining of pegRNA by two pseudoknot RNA motifs ("evopreQ1" or "mpknot"), and suggested that increased stabilities of epegRNAs underlie their better activities[19]. Therefore, we explored the stabilities of xrRNA-joined pegRNAs that had shown enhanced activities (see base conversion at *RIT1* in Fig. 3b, and small insertion at *EMX1* in Fig. 4a). First, we compared the degradation of in vitro-transcribed two groups of xr-pegRNAs and WT pegRNAs after incubation with nuclear extract from HEK293T cells (with endonuclease inhibited). Judged by the band intensities on the agarose gel, the xr-pegRNAs were significantly more resistant to degradation than the corresponding WT pegRNAs under this condition (Fig. 5a). More quantitative determinations of pegRNA and xr-pegRNA abundance in this in vitro system via RT-qPCR showed similar results (Supplementary Fig. 19a). As a positive control, pre-incubation of these RNA samples with an equivalent molar amount of Cas9 protein could lead to further protection of pegRNA and xr-pegRNA. In addition, based on RT-qPCR analysis of RNA samples from cells co-transfected with PE2 and WT pegRNA or xr-pegRNA plasmids, we found that the 3′-engineered xrRNA motif also increased the expression level of these pegRNAs in the cells (Supplementary Fig. 19b). These results confirm that 3′-joining of xrRNA to pegRNA can promote their stability.

In prime editing, pegRNA-templated reverse transcription is in competition with cellular nick processing/repair mechanisms. The 3′-stabilized xr-pegRNA would conceivably drive more efficient reverse transcription of the edit into a productive flap intermediate to enhance the overall PE efficiency. We next directly analyzed such PE intermediates, via an assay established in the epegRNA study[19]. HEK293T cells were co-transfected (24 h) with PE2 and WT pegRNA or xr-pegRNA plasmids for editing at the *RIT1* (+4G-to-A) and *EMX1* (+1CTG insertion) sites. The genomic DNA samples were prepared, followed by 3′-end oligo-dG labeling via terminal transferase. After site-specific amplifications, the compositions of such intermediates (flaps) were analyzed by NGS. In agreement with the mechanistic model of PE[1,19], the positions of the predominant 3′ flap-ends correlated to completed reverse transcription of the full-length RT template [and slightly beyond, i.e., +2 nt] (Fig. 5b and Supplementary Fig. 19c). Interestingly, besides the majority of such edit-containing intermediates, both sites also featured a proportion of unedited intermediates, most likely indicative of the competing repair processes. Of note, the use of xr-pegRNA was associated with an average of 2.7 ± 1.3-fold reduction in the proportion of

such unedited intermediates (Fig. 5b and Supplementary Fig. 19c, pie charts), consistent with a significant shift in nick-processing pathways, i.e., from nick repair to the intended reverse transcription of edits.

A stabilized pegRNA may possibly exhibit enhancements in both guide RNA activity and the precise, edit-installing efficiency. To separate these effects, the guide RNA activities for xr-pegRNAs were quantitated by a CRISPRa assay[19,20]. The reporters were constructed by inserting two respective target sequences directly upstream of a promoter-less (miniCMV) GFP cassette. Subsequently, the HEK293T cells were co-transfected with dCas9-vp64-p65-Rta (dCas9-VPR)[21], different WT pegRNAs or the xr-pegRNAs for 3-bp insertions, and the corresponding reporter plasmids. The levels of CRISPRa-activated GFP fluorescence (normalized to a transfection control of mCherry) were determined (Supplementary Fig. S19d). As positive controls, the conventional sgRNAs engaged more potent CRISPRa activities compared to either pegRNAs and xr-pegRNAs, likely attributed to their more compact and defined structure. However, the effects by xr-pegRNAs (*vs.* pegRNA) on CRISPRa activities did not correlate with their consistent impacts on PE efficiencies (Supplementary Fig. S19d, and see Fig. 4a for reference). These results suggest that under these tested conditions, the xrRNA-joining preferentially impact the prime editing function of pegRNA than its classic guide RNA activity, consistent with the proximity of the 3′-xrRNA to the edit-encoding sequence domain in pegRNA.

The xrRNA motifs are well-known to provide sub-genomic viral RNA with resistance to 5′-to-3′ exonucleases, owing to their mechanically stable ring-knot structure[15,22]. It is tempting to hypothesize that such a tertiary fold at the 3′ end of pegRNA may likewise contribute to degradation protection. To test this possibility, we introduced previously characterized mutations [U3C or C21G] in the xrRNA motifs to disrupt xrRNA tertiary folding[15,22]. We selected a site (*EMX1*) where the xr-pegRNAs had provided substantial greater PE activities compared to the pegRNA, and tested the effects by U3C- or C21G-mutated xr-pegRNAs on PE3-mediated base conversion (+1A-to-T) and small insertion (+1CTG). The results in HEK293T cells showed that both mutations substantially blunted the improvement effects by xr-pegRNA (Fig. 5c). Taken together, the above results support a model where the presence of a stably folded xrRNA motif at the 3′ of pegRNA protects the adjacent, edit-encoding sequence domain (i.e., the PBS and RT template), leading to enhanced PE activities.

**The xrPE platform shows comparable editing efficiency and fidelity as the epegRNA strategy.** Given the newly reported epegRNA strategy for PE improvements[19], we compared our xrPE system with its epegRNA counterpart (featuring an optimal Cr772 scaffold[23,24], 8-bp linker and tevopeQ1 motif). A total of nine different genetic modifications analyzed earlier in our study (consisting of base conversions, small deletions, or small insertions) were respectively targeted in HEK293T cells (Fig. 5d). In addition, we also compared the two PE platforms for three additional modifications analyzed previously in the epegRNA study[19] (Supplementary Fig. 20a). The results collectively demonstrated that the xrPE exhibited comparable editing efficiencies and edit:indel ratios as the epegRNA-based system (Fig. 5d and Supplementary Figs. 20a and 21a, b).

In our design, the xrRNA was directly joined to pegRNA without any linker sequences. As the epegRNA platform adopted an optimized linker between pegRNA and the 3′ evopreQ1/mpknot in an attempt to ensure structural flexibility[19], we also investigated the potential influence of linkers to the performances

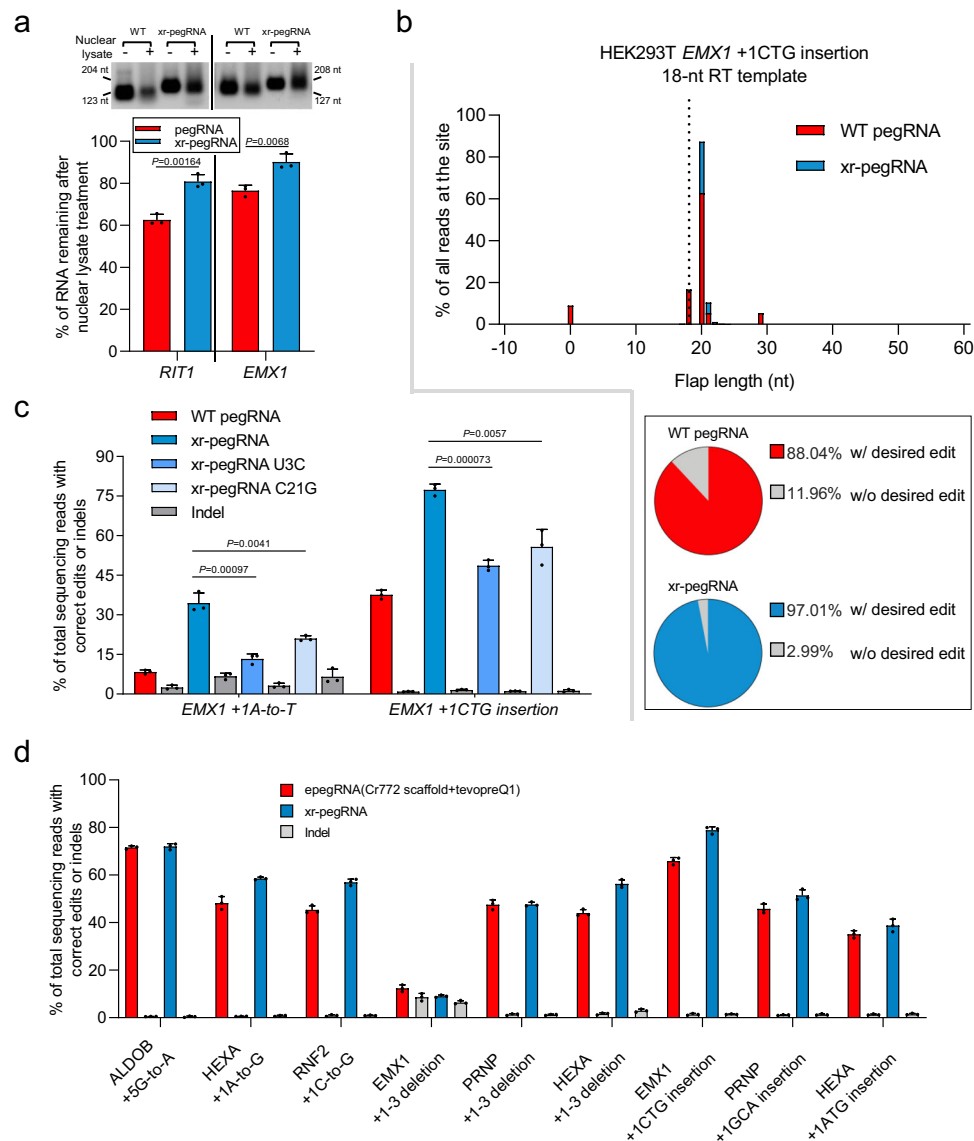

**Fig. 5 The 3′ xrRNA stabilizes xr-pegRNA to enhance prime editing, resulting in comparable performance as the latest epegRNA-based strategy.**
**a** The stability of in vitro-transcribed WT pegRNAs or the corresponding xr-pegRNAs upon exposure to HEK293T nuclear lysates. A representative agarose gel with samples from biological triplicates is shown. The sizes of the products (nt) are marked on the side. The levels for untreated WT pegRNAs or xr-pegRNAs were considered as 100%. Data presented are from quantitation of band intensity from 3 biological replicates (mean ± SD). The amounts of remaining pegRNAs or xr-pegRNAs were compared (two-tailed Student's *t* tests, with *P* values marked). **b** Comparison of PE intermediates generated by PE2 with either WT pegRNAs or xr-pegRNAs at *EMX1* site in HEK293T cells. The black dotted line represents the end of the full-length RT template (18 nt). In the histogram, the x axis corresponds to the sizes of the 3′ flaps, with the first base downstream of the PE2-induced nick denoted as position +1, while the y axis represents the relative abundance of the reads (percentage of all reads). The bottom box contains pie charts showing percentages of reads with (red/blue) or without (gray) intended edits. The data presented are calculated from an average of three independent biological replicates. **c** The WT pegRNA, xr-pegRNA, and two mutant xr-pegRNAs bearing either U3C or C21G mutations at the xrRNA domain were used in a PE3 context for base conversion or insertion at the *EMX1* sites in HEK293T cells. The editing efficiencies were determined (mean ± SD, *n* = 3 biological replicates) and compared (two-tailed Student's *t* tests, with *P* values marked). **d** The efficiencies by PE with xr-pegRNA (xrPE), or epegRNA (containing tevopreQ1 motif and Cr772 scaffold) for nine different genetic modifications attempted in our earlier experiments. Gray bars next to the red/blue-colored bars indicates the indel frequencies in association with a particular experiment group, where the editing efficiency is shown in red or blue (mean ± SD, *n* = 3 biological replicates). Source data are provided as a Source Data file.

by xrPE. To this end, computed linker sequences for 9 previously tested edits were inserted into corresponding xr-pegRNAs. The associated xrPE activities (for point mutations, 15-bp deletions or Flag tag insertions)[19] were analyzed in HEK293T cells (Supplementary Figs. 20b and 21c). Interestingly, the results showed that inclusion of linkers in xr-pegRNAs did not appear to affect the editing efficiencies, or the edit:indel ratios (Supplementary Figs. 20b and 21c). Similar observations were previously made

with one of the motifs (i.e., evopreQ1) used in epegRNA[19]. Although the structural determinants underlying the variable linker requirements for mpknot and xrRNA await to be elucidated, the unnecessity of a linker in the xr-pegRNA platform would support its convenient and predictable applications. Taken together, these comparative analyses demonstrate that our design of xrPE platform provides another readily applicable, and significantly improved tool for prime editing.

## Discussion

Precise genome editing provides opportunities to correct or rewire genetic information in cells, which holds great promise to revolutionize modern medicine and agriculture[25]. Although CRISPR/Cas9-induced DSB and the subsequent homology-directed repair can lead to installation of precise genetic modifications, such strategies are limited by DSB-associated genetic safety issues and the inefficiency of HDR[26]. The CRISPR-based cytosine base editors (CBEs) and adenine base editors (ABEs) could achieve permanent C-to-T and A-to-G conversions, respectively, without the requirement of DSBs[27,28]. The base editors have been subjected to extensive preclinical developments, which have shown promising results[29–31]. PE represents an exciting, later addition to the DSB-independent, precise genome editing toolbox. Capable of installing a wide variety of editing types, PE duly complements the current base editors for mutation scopes (beyond base transitions) and precision of editing (avoiding bystander mutations)[27,28], while featuring very low levels of off-target activities[1]. Nevertheless, the relatively low editing efficiencies of PE have presented a significant challenge to its wide applications.

The current study sought for an effective strategy to enhance prime editing. Aimed at protecting the 3′-end extension of pegRNAs that contain the intended editing information, several xrRNA motifs from the flaviviruses were appended downstream of pegRNAs (forming xr-pegRNA). These motifs have been known to protect the sub-genomic viral RNAs from 5′-to-3′ degradation[10]. Interestingly, here we show that the activities of the pegRNAs were generally enhanced by the joining of different viral xrRNAs at the 3′ end, yet to varied degrees (Figs. 1 and 2). Although the functional differences among various xrRNA groups appeared modest, it is interesting that the results from both PE2 and PE3 experiments showed similar patterns of relative performances by different viral motifs [both featuring a descending order of Zika, MVE, Dengue, MNV, and YF] (Fig. 2b, d). This is consistent with the notion that certain defined structural determinants appear to underlie the enhancement of pegRNA by 3′ xrRNA-joining (see discussions below). Such corroborative patterns from independent comparisons also strongly support the hitherto best-performing Zika xrRNA motif as an effective pegRNA potentiator, which we subsequently adopted for constructing xr-pegRNAs throughout the study (to establish the xrPE platform).

Further characterizations in multiple cell types indicate that xrPE provides notable enhancements over canonical PE3 for precise base conversions, small deletions and small insertions (Figs. 3 and 4), while featuring undiminished on-target edit:indel ratios and largely undetectable off-target editing (Supplementary Figs. 8, 10, and 12–16). Interestingly, the enhancement effects by the use of xr-pegRNAs appear more pronounced as they were introduced via lentiviral vectors (Supplementary Figs. 17 and 18, see Figs. 3b and 4a for transfection conditions). These results demonstrate another common setting, besides transfections, for future xrPE application. Additionally, as lentiviral vectors generally drive less robust payload expression than effective transfections, such data may implicate a further advantage by xrPE under many conditions where pegRNA expression may be limited.

Our initial investigations were carried out independently from a very recent development of epegRNA system by others, which reported a strategy that featured 3′-addition of other structured RNA motifs to stabilize pegRNA and improve PE[19]. Additional experiments through the revision of the present work also confirm the increased stability of xr-pegRNAs (vs. pegRNA), and suggest a mechanism where 3′-stabilized xr-pegRNAs act to enhance productive reverse transcription of intended edits

(Fig. 5a, b and Supplementary Fig. 19a–d). Importantly, direct comparison of PE efficiencies showed largely similar performances by xrPE and the epegRNA system (Fig. 5d and Supplementary Figs. 20 and 21). Therefore, the present xrPE system complements the newly emerged epegRNA system, highlighting the effect by optimized pegRNA architecture on PE enhancements. Regarding prime editing accuracy, the xrPE system features overall similar edit:indel ratios as PE3 (Supplementary Figs. 8, 10d, 12, and 13c), and as the epegRNA strategy (Supplementary Figs. 21a, b). These observations support a model that the majority of PE3-related undesired indels are directly coupled with the generation of the editing intermediate (3′ flaps). As a major goal for precise genome editing is to minimize such undesired indels, a better understanding to the cellular responses down-stream of the reversely transcribed 3′ flap intermediates may further contribute to development of advanced PE methods.

On the other hand, the increased sizes for xr-pegRNAs may require careful consideration, especially in delivering formats other than plasmids or viral vectors (e.g., mRNA/protein PE). Currently, chemical synthesis of RNA oligonucleotides beyond the usual lengths of canonical pegRNAs remain challenging. The relative bulkiness of the added xrRNA motif (>70 nt) presents a further burden in this regard. The alternative use of in vitro-transcribed xr-pegRNAs may potentially be implemented in mRNA/protein PE applications. However, further examinations would be required to determine whether the benefit of adopting the polymerase-dependent, larger xr-pegRNAs (vs. pegRNAs) could sufficiently outweigh their lack of stabilizing chemical modifications.

Most xrRNA-related studies have focused on their resistance against 5′ exonucleases[10,11,14]. Interestingly, these structures have shown high levels of intrinsic mechanical strength[15]. Such mechanical property may contribute to their ability to also mediate 3′ protection of pegRNAs. Indeed, two mutant, rigidity-reduced Zika xrRNA motifs (as the 3′ domain of xr-pegRNA) showed substantially weakened effects on enhancing PE (Fig. 5c). Along the same line, it is possible that variable mechanical properties associated with different virus-derived xrRNA motifs, determined by their diverse primary sequences[22,32], may underlie their quantitatively differential enhancement effects on pegRNA (see Fig. 2b, d). We believe that future explorations of intrinsically stable RNA motifs (natural and engineered) may prove fruitful for identification of additional pegRNA enhancers. Besides their high adaptability, one could envision that such autonomously stable RNA motifs may also present advantages to operate independent of the cellular contexts to support a wide range of PE applications.

In summary, our studies establish xrPE as a significantly improved and highly adaptable prime editing platform, which represents an important addition to the precise genome editing toolbox for future development and applications.

## Methods

**Plasmid construction.** pCMV-PE2 plasmid was purchased from Addgene (Addgene, #132775). For expression of the WT pegRNA, the plasmid backbone was amplified from pGL3-U6-sgRNA-EGFP (Addgene, 107721) using Phanta® Max Super Fidelity DNA Polymerase (Vazyme) for removal of the original sgRNA scaffold (Supplementary Table 1). The resulting plasmid was cut by BsaI-HFv2 (NEB) for overhangs. The pegRNA scaffold oligos (featuring compatible overhangs) (Supplementary Table 1), spacer oligos (top strand with ends of 5′ ACCG and 3′ GTTTT, bottom strand with 5′ CTCTAAAAC overhang), and pegRNA 3′ extension oligos (top strand with 5′ GTGC overhang, bottom strand with 5′ AAAA overhang) were synthesized and annealed. The annealed scaffold fragment was phosphorylated. Finally, four fragments (annealed spacer, annealed 3′ extension, phosphorylated scaffold, and the cut backbone) were assembled by DNA ligase. The sequence information for pegRNAs and nick-sgRNAs is provided in Supplementary Data 1. Assembled plasmids were transformed into E. coli and screened using Ampicillin. Based on the pegRNA construct, the joining of xrRNA

sequence was carried out using recombinase-based cloning (Vazyme, ClonExpress II One Step Cloning Kit, #C112-02-AB).

**Cell culture, transfection, and harvest**. HEK293T (ATCC CRL-3216), HeLa (ATCC CCL-2), and Neuro-2a (N2a, ATCC HTB-96) cells were cultured in Dulbecco's Modified Eagle Medium (Gibco) supplemented with 10% fetal bovine serum (FBS) (v/v) (Gemini) and incubated at 37 °C with 5% CO2. None of the cell lines used are listed in the ICLAC database. For plasmid transfection, cells were seeded on poly-D-lysine-coated 24-well plates and transfected at ~70% confluence using EZ Trans (Shanghai Life iLab Biotech Co., Ltd), according to the manufacturer's protocols. For PE2, 900 ng pCMV-PE2 plasmid, together with 300 ng pegRNA plasmid (with an EGFP marker) were transfected into cells per well. For PE3, 900 ng pCMV-PE2 plasmid, together with 300 ng pegRNA plasmid and 100 ng nick-sgRNA plasmid were transfected into cells per well. 72 h after transfection, the cells were subjected to Fluorescence Activated Cell Sorting (FACS) to harvest EGFP+ cells for sequencing analyses. In the initial experiments to analyze the editing of a fluorescent reporter plasmid (20 ng), a modified pegRNA plasmid without fluorescent marker was used. The reporter-edited cells were subjected to flow cytometry using BD LSRFortessa. The data were analyzed using FlowJo (X 10.07r2). The percentage of EGFP expression marked the prime editing efficiency, which was calculated as: comp-GFP-A+/comp-DsRed-A+.

**Genomic DNA extraction and genotyping**. The genomic DNA of GFP+ cells was extracted using QuickExtract™ DNA Extraction Solution (Lucigen) according to manufacturer's protocols. The isolated DNA was PCR-amplified with Phanta® Max Super-Fidelity DNA Polymerase (Vazyme). Primers used are listed in Supplementary Tables 2 and 3.

**Targeted deep-sequencing**. Target sites were amplified from extracted genomic DNA using Phanta® Max Super-Fidelity DNA Polymerase (Vazyme). PCR products with different barcodes were pooled together for deep sequencing on Illumina HiSeq X Ten platform (2 × 150 PE) by Annoroad Gene Technology (Beijing, China). Different experimental conditions were differentiated by bar codes and experimental repetitions were included in different pools. Sequencing reads were demultiplexed using AdapterRemoval (version 2.2.2), and the pair-end reads with 11 bp or more alignments were combined into a single consensus read. All processed reads were then mapped to the target sequences using the BWA-MEM algorithm (BWA v0.7.16). Prime editing efficiency was calculated as: percentage of (number of reads with the desired edit that do not contain indels)/ (total mapped reads). Indel frequency was calculated as: number of indel-containing reads/total mapped reads. Mutation rate was calculated using bam-readcount with parameters -q 20 -b 30 -i.

**Western blotting**. For Western blotting, 24-well plate HEK293T cells were lysed by RIPA. The antibodies used included anti-Cas9 (Genscript (A01935, clone 4A1), 1:500), anti-GAPDH (Santa cruz (sc47724, clone 0411), 1:1000), and anti-GFP (ABclonal (AE012), 1:2000). Images were captured with Amersham Imager 600. Uncropped blots for the presented results are provided in the Source Data file.

**Off-target analysis**. Potential off-target sites were predicted in the human genome (GRCh38/hg38) with Cas-OFFinder (2.4) (http://www.rgenome.net/cas-offinder). The sequences around the predicted off-target sites were amplified using Phanta Max Super-Fidelity DNA Polymerase (Vazyme), and subjected to high-throughput sequencing with the Illumina HiSeq X Ten (2 × 150 PE) at Annoroad Gene Technology, Beijing, China. The amplicons were analyzed with as the method described in Targeted deep-sequencing and the off-target sites are listed in Supplementary Table 4. Primers used are listed in Supplementary Table 5.

**Lentiviral vector-transduced xr-pegRNA on editing with PE2 and PE3**. Lentiviral transfer plasmids contained a U6 promoter driving the expression of WT pegRNA or xr-pegRNA, and a mCherry–P2A–Puro marker under the EF1α core promoter. For packaging, HEK293T cells were seeded on six-well plates at $6.5 \times 10^5$ cells per well in DMEM supplemented with 10% FBS. At 12–16 h after seeding (70–90% confluency), cells were transfected with a mix of the core pegRNA plasmid (1 μg), pMD2.G (0.6 μg, Addgene no. 12259) and psPAX2 (1 μg, Addgene no. 12260), together in 7.8 μl of EZ Trans reagent (Shanghai Life iLab Biotech Co., Ltd). 4~6 h after transfection, the cells were re-fed with fresh medium. The supernatants were collected 48 hours after transfection. Following cellular debris removal by centrifugation (1000×g for 8 min), the supernatants were filtered through a 0.45-μm filter and stored at −80 °C. For transduction of pegRNAs or xr-pegRNAs, $2 \times 10^5$ HEK293T cells in regular culture medium were added with 20 μl of the lentiviral vector-containing supernatant in 12-well plates. 12 h after the initial infection, the culture medium was replaced with fresh medium. After 6 days, the transduced HEK293T cells were re-seeded on 24-well plates $1 \times 10^5$ cells per well. 16 h after seeding, cells were transfected at 60–80% confluency with 900 ng of EF1a–PE2 plasmid (PE2) in 2.7 μl of transfection reagent, or with a mix of 900 ng of EF1a–PE2 and 100 ng nick-sgRNA plasmid (PE3) in 3 μl of transfection reagent. GFP positive cells were sorted 4 days after transfection, and subjected to genomic DNA preparation.

**RT–qPCR of pegRNAs**. Transfection of HEK293T with PE2 plasmids (with xr-pegRNA or pegRNA) was performed as described above. Total RNA from transfected cells was isolated using the RNA isolater Total RNA Extraction Reagent (Vazyme). The HiScript Q RT SuperMix for qPCR (+gDNA wiper) (Vazyme) was used to generate cDNA using random hexamers. The qPCR was carried out with primers for pegRNA scaffold using a commercial reaction mix from Vazyme (AceQ qPCR SYBR Green Master Mix [Low ROX Premixed]). The pegRNA signal was normalized to the PE2 mRNA signal for transfection efficiency. Fold changes in abundance were calculated using the 2-ΔΔCt method. Primer sequences are available in Supplementary Table 6.

**In vitro pegRNA stability assay**. The pegRNAs or xr-pegRNAs were in vitro transcribed from T7 promoter-led templates using the MEGAshortscript™ T7 kit (Invitrogen). Their degradation profile under an endonuclease-inhibited condition was analyzed as recently described[19], with minor modifications. Briefly, the nuclear extracts were prepared from near-confluent HEK293T cells using the ExKine™ kit (Abbkine). One μg of in vitro-transcribed RNA was added to each sample that contained ±1.5 μl of fresh nuclear lysate in a total volume of 10 μl of reaction buffer (20 mM Tris-HCl, pH 7.5, 5 mM MgCl2, 50 mM NaCl, 2 mM DTT, 1 mM NTP and 0.8 U μl−1 of RNaseOUT [from Vazyme] endonucleases inhibitor). After incubation at 37 °C for 20 min, the samples were resolved on 2.0% agarose gels and stained with Ultra GelRed (Vazyme). The intensities of the bands were quantitated by ImageJ (1.53i). The uncropped pictures of the gels are provided in the Source Data file. As a positive control of degradation protection, 1 μg of RNA samples were pre-incubated with (or without) 5 μg Cas9 protein at the room temperature for 10 min. The samples were next subjected to incubation with 3 μl of nuclear lysate in the same condition as above. In all, 2 μl 10 U μl−1 of protease K solution was then added to terminate the reaction. The remaining RNA was precipitated using isopropyl alcohol and subjected to RT–qPCR analyses.

**CRISPRa transcriptional activation assay**. In all, $5 \times 10^4$ HEK293T cells were transfected at approximately 60~90% confluency with 1 μl of EZ Trans, CMV-dCas9-VPR (300 ng), targeted EGFP reporters (30 ng) and different guide RNAs (sgRNA, pegRNA, xr-pegRNA, or a control sgRNA, 100 ng). A 10 ng CMV-mCherry plasmid was also included to label transfected cells. After 2 days, cells were subjected to flow cytometry. The EGFP mean fluorescence intensity (MFI) was normalized to that of mCherry to represent corresponding CRISPRa activities.

**Analyses for nicked PE intermediates**. HEK293T cells in 24-well plates were transfected with PE2 and pegRNAs/xr-pegRNAs (900 ng and 300 ng, respectively). After 24 h, genomic DNA was isolated from the cells using the Tianamp Genomic DNA Kit. Then, the 3′ termini were tailed using Terminal Deoxynucleotidyl Transferase (yeasen) and dGTP, according to the manufacturer's instructions. After a step of sample purification (AxyPrep PCR clean kit), the labeled sites were amplified using a site-specific forward primer and an oligo-C (C18) reverse primer for subsequent NGS analyses.

**Data analyses**. All data presented were based on three biological replicates. Analyses and graphing were carried out with GraphPad Prism (version 8). Data are presented as means ± SD (or ±SEM) as indicated in the legends. In box plots, the center line shows medians and the box limits correspond to upper the lower quartiles. In violin plots, the thicker dotted line shows medians, while the thinner dotted lines correspond to quartiles. Statistical significance of differences between two groups was determined using Student's t tests (unpaired, unless indicated specifically). When multiple t tests were performed in parallel for larger numbers of comparisons (see Figs. 3 and 4), discoveries were determined using the two-stage linear step-up procedure of Benjamini, Krieger, and Yekutieli[33], with Q = 1%.

**Reporting summary**. Further information on research design is available in the Nature Research Reporting Summary linked to this article.

## Data availability
Targeted amplicon sequencing data has been deposited in the NCBI-SRA under BioProject number PRJNA761932 (https://www.ncbi.nlm.nih.gov/bioproject/PRJNA761932/). Source data are provided with this paper.

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

## Acknowledgements

This work is supported by grants from the National Key Research and Development Program of China (2021YFF1000704 and 2019YFA0802802 to J.L.), the Emergency Key Program of Guangzhou Laboratory (EKPG21-18 to X.H.), and the Leading Talents of Guangdong Province Program (608285568031 to X.H.). We thank members of Huang lab and Wang lab and Liu lab for helpful discussions. We thank the Molecular and Cell Biology Core Facility (MCBCF) at the School of Life Science and Technology, ShanghaiTech University for providing technical support.

## Author contributions

X.W., X.H., and J.L. conceived the study and designed experiments. G.Z., Y.L., S.Q., and D.C. performed the experiments with the assistance of Y.Y. and Q.J., and colleagues of Molecular and Cell Biology Core Facility (MCBCF) at the School of Life Science and Technology of ShanghaiTech University. S.H. analyzed the data. All authors discussed the results and approved the manuscript. G.Z., Y.L., X.W., X.H., and J.L. wrote and revised the manuscript.

## Competing interests

The authors declare no competing interests.
