## [Peer Review File · Nature Communications]

Reviewers' Comments:

Reviewer #1:

Remarks to the Author:

Reviewer's comments:

In this manuscript, Zhang et al. describe the xrPE system they developed by applying pegRNA using a virus-derived xrRNA motif to increase the editing efficiency of the prime editor. This xrPE system demonstrates improved editing efficiency compared to the existing canonical PE3 system, and it is believed that it can be used in several applications. However, since the length of xrRNA is longer than that of the existing pegRNA, so its suitability for application to mRNA or protein PE systems is questionable. In addition, I have several major concerns regarding this manuscript.

Major concerns:

1. The authors attempted to increase editing efficiency by applying five different xrRNA motifs derived from viruses to pegRNAs in the prime editor system. When applied to the PE system with five different xrRNA motifs derived from viruses, was there no difference in unwanted mutations such as indels, other than the desired exact corrections? Reducing the introduction of unwanted mutations is also one of the challenges that the PE system has to overcome. Furthermore, authors should present desired as well as undesired mutations in data.
2. Nelson et al. reported an engineered pegRNA containing a structured RNA motif 3' in PBS to prevent degradation of the pegRNA. The authors should compare the xrPE system with the PE system using epegRNA.
3. The xrPE system can be easily applied to plasmids, but it may be difficult to apply long xrRNAs to editing methods using an mRNA or RNP PE system. The authors should treat the xrRNA with either mRNA PE or protein PE to demonstrate its versatile applicability.
4. In the experimental group of the xrPE system using engineered xrRNA, more undesirable mutations such as indels were found than in PE3. Why was this the case? The authors should also present direct evidence that the stability of xrRNA is increased.

Reviewer #2:

Remarks to the Author:

Prime editing is a new CRISPR-based genome editing tool that can install precise changes in the genome without the requirement for double-strand breaks or exogenous donor DNA templates; however, its utility is hampered by low and inconsistent efficiency. Zhang et al posit that appending RNA-stabilizing motifs to pegRNAs may increase prime editing efficiency by enhancing pegRNA stability. Specifically, the authors show that appending xrRNA sequences from 4 different viruses enhances prime editing activity. Focusing on Zika virus xr-pegRNAs, the authors show consistent improvements in prime editing efficiency in three cell types for a variety of edit types without increasing byproducts and off-target editing.

This paper is straightforward and convincing. Its impact is dampened by a recent report from Nelson et al that performs a more comprehensive screen of RNA-stabilizing motifs, replicating the result of broadly increased prime editing efficiency but with more extensive mechanistic analysis and more extensive proof of increased utility. Nonetheless, independent validation of this concept is worthwhile for this fast-moving field.

I would suggest the following improvements to the manuscript:

Major points

1. In all summaries of the relative efficiency of xrRNA-joined pegRNAs and standard pegRNAs in the manuscript and figures, the authors should state both median and mean editing foldchange. Mean can be skewed by outlier data points, so median is a more fair way of representing the expected enhancement at a median target site.
2. The authors have only used a single mode of introducing prime editor and pegRNA into cells, transient transfection. Stable lentiviral transduction is a common method of editing used in genetic screening. It is unclear if xrRNA-joined pegRNAs would have an effect in this context, as the inferred mechanism of increasing RNA stability may not have the same importance. The authors should test xrPE2 in a lentiviral context to address whether there is utility for xrPE2/3 in this common setting.

3. The authors infer that enhanced editing efficiency derives from enhanced pegRNA stability. This seems a plausible mechanism, especially given the published data from Nelson et al that show this. However, there is no mechanistic data in this manuscript. The authors should perform at least cursory mechanistic analysis of the degree of enhanced stability provided by xrRNA attachment to address this presumed mechanism.

4. In Nelson et al, effort is made to identify non-interfering nucleotide linkers between the pegRNA and RNA-stabilizing motif. The authors should discuss this idea in context of their findings. Does their data give clues as to whether this consideration is also relevant for xrPE2/3?

Minor points

1. The authors originally tried a number of xrRNA sequences, which vary in efficacy. In the discussion, could they address why certain xrRNA sequences might be more effective than others?

2. Line 109-110, correlates well might be an over statement since Zika shows higher in flow but less in Western

3. The authors may want to explain better why they chose to follow up with Zika xrRNA since the plots in Fig. 2 don't make this obvious.

Below are our point-by-point replies:

Reviewer #1 (Remarks to the Author):

Reviewer's comments:

In this manuscript, Zhang et al. describe the xrPE system they developed by applying pegRNA using a virus-derived xrRNA motif to increase the editing efficiency of the prime editor. This xrPE system demonstrates improved editing efficiency compared to the existing canonical PE3 system, and it is believed that it can be used in several applications. However, since the length of xrRNA is longer than that of the existing pegRNA, so its suitability for application to mRNA or protein PE systems is questionable.

[Authors' response]

We thank the reviewer very much for this important comment regarding the potential application of xr-pegRNA in mRNA or protein PE. The increased sizes of xr-pegRNAs are indeed a burden for chemical synthesis. We have added related comments to the Discussion (line 422-426).

The alternative use of *in vitro*-transcribed xr-pegRNAs may potentially be considered in mRNA or protein PE applications. The revised manuscript contains additional data that xr-pegRNAs can be prepared readily with *in vitro* transcription (current Fig. 5a, note their bigger sizes compared to the pegRNAs seen on the agarose gel).

However, further investigations would be required to determine whether the benefit of adopting the polymerase-dependent, larger xr-pegRNAs (*vs.* pegRNAs) could sufficiently outweigh their lack of stabilizing chemical modifications. This comment is also added to the Discussion (line 426-430).

In addition, I have several major concerns regarding this manuscript.

Major concerns:

1. The authors attempted to increase editing efficiency by applying five different xrRNA motifs derived from viruses to pegRNAs in the prime editor system. When applied to the PE system with five different xrRNA motifs derived from viruses, was there no difference in unwanted mutations such as indels, other than the desired exact corrections? Reducing the introduction of unwanted mutations is also one of the challenges that the PE system has to overcome. Furthermore, authors should present desired as well as undesired mutations in data.

[Authors' response]

We thank the reviewer very much for this question. We apologize for having not presented the unwanted indel data for our initial characterization of different xr-pegRNA constructs. Such data are now presented in the form of edit:indel ratios within current Supplementary Fig. 3d (the reporter target, PE2) and Supplementary 6 (genomic

targets, PE2 and PE3). The ratios were respectively determined for every editing sample and were subsequently averaged for each target site. For the entire manuscript, we have also supplied data of edit:indel ratios for most experiments to facilitate the comparisons of editing accuracies, in addition to the already presented aspect of efficiencies, between xrPE and other PE applications. These data have been added within supplementary figures.

In Supplementary Fig. 3d, an overall enhanced (except for one construct) edit:indel ratios by different xr-pegRNA constructs (with PE2) on the reporter target were observed (line 116-119). This pattern correlates with that for the editing efficiencies (compare Supplementary Fig. 3c and 3d). Indeed, the indel rates generally did not increase with the xr-pegRNA-enhanced editing rates for this plasmid-borne target (ranging between 99-107% of the levels for the WT pegRNA group, see accompanied source data).

In Supplementary Fig. 6, due to the relatively low levels of indel reads for genomic targets, some variations among replicates are observed for certain experimental groups. Nevertheless, an overall trend of similar or somewhat increased edit:indel ratios were observed for different xr-pegRNA constructs in both PE2 and PE3 formats (Supplementary Fig. 6a, b, line 136-137, 149-150).

A further summary in the context of PE3 is provided in Supplementary Fig. 6c, where the edit:indel ratio data from all sites were grouped together. Here, different viral xrRNA constructs are all associated with median edit:indel levels ≥ 1.0 [relative to the pegRNA group] (Supplementary Fig. 6c). These results collectively demonstrate undiminished PE accuracies by the use of different xr-pegRNA constructs. Additionally, the differences among the use of different xrRNA are not apparent (line 163-166).

2. Nelson et al. reported an engineered pegRNA containing a structured RNA motif 3' in PBS to prevent degradation of the pegRNA. The authors should compare the xrPE system with the PE system using epegRNA.

[Authors' response]

We thank the reviewer very much for suggesting the comparison of our xrPE with the newly developed epegRNA-based PE platform ¹, and for pointing out the converging principles of design. As the epegRNA study was published just prior to the initial submission of our work, the original manuscript did not contain experimental comparisons. We have now duly performed direct comparisons between the two systems. The data are included in current Fig. 5d and Supplementary Fig. 20, 21.

An optimized epegRNA configuration (with a Cr772 scaffold, 8-bp linker and a tevopeQ1 motif) was used for the comparisons. We chose to apply xrPE and epegRNA PE to a first set of 9 edits tested earlier in our study (Fig. 5d and Supplementary 21a), and a second set of 3 edits reported in the epegRNA study (Supplementary Fig. 20a, 21b). These edits involve various base conversions, small insertions and small deletions. Notably, the results showed that the xrPE had very similar performances as the epegRNA system in both editing efficiencies and indel rates (line 344-351). Therefore,

the present xrPE system complements the newly emerged epegRNA system, highlighting the effect by optimized pegRNA architecture on PE enhancements.

3. The xrPE system can be easily applied to plasmids, but it may be difficult to apply long xrRNAs to editing methods using an mRNA or RNP PE system. The authors should treat the xrRNA with either mRNA PE or protein PE to demonstrate its versatile applicability.

[Authors' response]

We thank the reviewer again for this important point. Currently, due to issues of size limits, chemical syntheses of the xr-pegRNAs are indeed challenging. Such comments are added to the discussion (line 422-426).

Alternatively, *in vitro*-transcribed xr-pegRNA may be adopted in mRNA/protein PE experiments. Unfortunately, despite numerous attempts, we did not detect PE activities via electroporation of either protein or mRNA PE2 together with the purified xr-pegRNA in HEK293T cells, most likely due to our lack of expertise in PE2 mRNA/protein preparation. In contrast, co-electroporation of Cas9 protein with xr-pegRNA led to robust target cleavage, indicating successful delivery of xr-pegRNA (data not shown in the manuscript, see below).

We sincerely apologize for our inability to provide data regarding xr-pegRNA in mRNA or protein PE. As such experiments apparently require special expertise², we therefore plead for the understandings by the reviewers. Additionally, whether the benefit of adopting the polymerase-dependent, larger xr-pegRNAs (*vs.* pegRNAs) could sufficiently outweigh their lack of stabilizing chemical modifications requires careful consideration in the future (added to Discussion, line 426-430).

On the other hand, we now have shown in the revised manuscript (as requested by reviewer #2) that there is significant enhancement by the lentiviral vector-delivered xr-pegRNA (*vs.* pegRNA, current Supplementary Fig. 17, 18, line 258-276) for PE

activities. These results demonstrate the applicability of xr-pegRNA for another common delivery system, besides plasmid transfections.

4. In the experimental group of the xrPE system using engineered xrRNA, more undesirable mutations such as indels were found than in PE3. Why was this the case?

[Authors' response]

We thank the reviewer very much for this important question. Indeed, when all data related to the indels are considered, the application of xrPE are generally associated with similar edit:indel ratios as PE3 (current Supplementary Fig. 8, 10d, 12, 13c), indicating an overall trend of proportionally increased indel rates. To be informative, the comparative patterns for edit:indel ratios by xrPE and PE3 are also clearly described throughout the revised text. The trend for indel rates is also summarized in the current Supplementary Fig. 14 (original Fig. S11). The relevant text section is modified to better describe such a trend (line 229-239).

Interestingly, the epegRNA-based PE system also showed such a feature¹. These corroborative observations suggest that the majority of these undesired indels are directly coupled with the generation of the editing intermediate (3' flaps) by PE. As a major goal for precise genome editing is to minimize such undesired indels, a better understanding to the cellular responses down-stream of the reversely transcribed 3' flap intermediates may further contribute to development of advanced PE methods. Such comments are added to the Discussion (line 415-421).

The authors should also present direct evidence that the stability of xrRNA is increased.

[Authors' response]

We thank the reviewer very much for this comment. We have performed additional experiments to directly analyze the mechanisms whereby addition of 3'-xrRNA motif to pegRNA may enhance prime editing (line 279-341).

Firstly, the *in vitro*-transcribed xr-pegRNA and pegRNA were treated with nuclear lysates from HEK293T cells under an endonuclease-inhibited condition (current Fig. 5a and Supplementary Fig. 19a). The levels of pegRNA were reduced more substantially than those of xr-pegRNA, validating that xr-pegRNA was more resistant to degradation by exonuclease activities. In correlation, the levels of these xr-pegRNAs in the cells were also higher than their corresponding pegRNAs, judged by qPCR analyses (current Supplementary Fig. 19b).

Next, we analyzed the compositions of nicked PE intermediates associated with the use of the xr-pegRNA and pegRNA (current Fig. 5b and Supplementary 19c), by an assay established in the epegRNA study¹. Indeed, most of such intermediates contained desired edits, corresponding to reverse transcription of the RT template. On the other hand, some nicked species were found to not contain the desired edits, and were likely to represent certain nick processing/repair intermediates. Interestingly, the use of xr-

pegRNA corresponds to substantial (2.7 ± 1.3 fold) reduction in the proportion of such unedited intermediates. These results suggest that 3'-xrRNA-protected pegRNA can lead to a shift in nick-dependent responses, i.e., toward productive reverse transcription of intended edits.

Furthermore, the importance of the xrRNA tertiary fold to the enhanced PE activity by the xr-pegRNA was validated by the use of structure-disrupting mutant xrRNA domains (current Fig. 5c). We also supplemented data showing that the canonical guide RNA activities by given xr-pegRNAs (in CRISPRa assays) do not necessarily correlate with those for PE (current Supplementary Fig. 19d). Therefore, the prime editing activity is selectively enhanced in xr-pegRNA, consistent with the placement of the xrRNA at the 3'-end of pegRNA (adjacent to the edit-encoding sequence domain).

Reviewer #2 (Remarks to the Author):

Prime editing is a new CRISPR-based genome editing tool that can install precise changes in the genome without the requirement for double-strand breaks or exogenous donor DNA templates; however, its utility is hampered by low and inconsistent efficiency. Zhang et al posit that appending RNA-stabilizing motifs to pegRNAs may increase prime editing efficiency by enhancing pegRNA stability. Specifically, the authors show that appending xrRNA sequences from 4 different viruses enhances prime editing activity. Focusing on Zika virus xr-pegRNAs, the authors show consistent improvements in prime editing efficiency in three cell types for a variety of edit types without increasing byproducts and off-target editing.

This paper is straightforward and convincing. Its impact is dampened by a recent report from Nelson et al that performs a more comprehensive screen of RNA-stabilizing motifs, replicating the result of broadly increased prime editing efficiency but with more extensive mechanistic analysis and more extensive proof of increased utility. Nonetheless, independent validation of this concept is worthwhile for this fast-moving field.

I would suggest the following improvements to the manuscript:

Major points

1. In all summaries of the relative efficiency of xrRNA-joined pegRNAs and standard pegRNAs in the manuscript and figures, the authors should state both median and mean editing foldchange. Mean can be skewed by outlier data points, so median is a more fair way of representing the expected enhancement at a median target site.

[Authors' response]

We thank the reviewer very much for this suggestion. We certainly concur with the reviewer that as PE efficiencies at different sites significantly vary, the presentation of median levels of enhancement by xr-pegRNA is important. In the revised manuscript, we now describe the fold-change by xr-pegRNAs (xrPE) over the canonical PE systems by a format of “median (mean)” throughout the report (related to Fig. 2b, d, 3c, e, 4b,

d, and Supplementary Fig. 10b, 13b, 17b, 18b). Such a format is initially introduced in line 138-139 of the revised text.

2. The authors have only used a single mode of introducing prime editor and pegRNA into cells, transient transfection. Stable lentiviral transduction is a common method of editing used in genetic screening. It is unclear if xrRNA-joined pegRNAs would have an effect in this context, as the inferred mechanism of increasing RNA stability may not have the same importance. The authors should test xrPE2 in a lentiviral context to address whether there is utility for xrPE2/3 in this common setting.

[Authors' response]

We thank the reviewer very much for this suggestion. We performed additional experiments with lentiviral vector (LV)-introduced xr-pegRNAs (current Supplementary Fig. 17, 18, line 258-276). The LV-transduced cells were further transfected with plasmids for either PE2 and PE3. The results demonstrate that LV-introduced xr-pegRNAs drove substantially higher levels of prime editing than the original pegRNAs. As lentiviral vectors generally drive less robust payload expression than effective transfections, such pronounced levels of enhancement also implicate a further advantage by xrPE under conditions where pegRNA expression may be limited (discussion added in line 399-405).

3. The authors infer that enhanced editing efficiency derives from enhanced pegRNA stability. This seems a plausible mechanism, especially given the published data from Nelson et al that show this. However, there is no mechanistic data in this manuscript. The authors should perform at least cursory mechanistic analysis of the degree of enhanced stability provided by xrRNA attachment to address this presumed mechanism.

[Authors' response]

We thank the reviewer very much for this important question, which was also raised by reviewer #1. We have duly performed additional experiments to gain some mechanistic insights to the improved PE activity by xr-pegRNAs. Please refer to our response to reviewer #1 for details (corresponding to line 279-341 of the revised manuscript). In summary, we provided additional data to show that xr-pegRNAs are more protected from degradation when exposed to nuclear lysates, which correlates with their increased levels upon transfection (current Fig. 5a and Supplementary Fig. 19a, b). Capture-and-sequencing of nicked intermediates at target sites showed that the use of xr-pegRNA was associated with notably reduced proportions of unedited intermediates (current Fig. 5b and Supplementary Fig. 19c). Moreover, structure-disrupted mutants of xrRNA showed severely diminished editing-enhancement effects on pegRNAs (current Fig. 5c). Taken together, these added results are consistent with a mechanism where the 3'-stabilized xr-pegRNAs act to enhance productive reverse transcription of intended edits.

4. In Nelson et al, effort is made to identify non-interfering nucleotide linkers between the pegRNA and RNA-stabilizing motif. The authors should discuss this idea in context of their findings. Does their data give clues as to whether this consideration is also relevant for xrPE2/3?

[Authors' response]

We thank the reviewer very much for this question. Through our development of xrPE, we did not include linker sequences upstream of the xrRNA motif. It is indeed interesting that in the epegRNA report, the implementation of linkers showed some benefit for the epegRNAs featuring one (mpknot), but not the other (evopreQ1) RNA motifs¹. We also understand the sound logic behind their development of a tool for *in silico* linker optimization (to minimize interference with the pegRNAs).

To examine the potential influence by an optimized linker on xr-pegRNA performance, the computed linker sequences for 9 edits (analyzed by Nelson et. al.,) were inserted into corresponding xr-pegRNAs. They were compared in parallel with the non-linker version of xr-pegRNAs for priming editing activities. Interestingly, our results showed that the inclusion of linker sequences in xr-pegRNAs did not affect the editing efficiencies or edit:indel ratios (current Supplementary Fig. 20b, 21c, line 352-360).

The epegRNA study suggested the relative larger size of mpknot for the selective benefit of linkers for constructs containing this motif. However, the Zika xrRNA motif (81-nt) used here is bigger than the mpknot (55-nt). Therefore, the structural determinants underlying the variable linker requirements for mpknot and xrRNA await to be elucidated. On the other hand, the unnecessary of a linker in the xr-pegRNA platform would support its convenient and predictable applications. Such comments are added in line 360-363 of the Results.

Minor points

1. The authors originally tried a number of xrRNA sequences, which vary in efficacy. In the discussion, could they address why certain xrRNA sequences might be more effective than others?

[Authors' response]

We thank the reviewer very much for this question. Our major characterizations of different xrRNA motifs were based on PE2 and PE3 results at 6 genomic sites.

Indeed, the results showed that the activities of the pegRNAs were generally enhanced by the joining of different viral xrRNAs at the 3' end, although to varied degrees (see Fig. 1, 2). Although the functional differences among various xrRNA groups appeared modest, it is interesting that the results from both PE2 and PE3 experiments showed similar patterns of relative performances by different viral motifs [both featuring a descending order of Zika, MVE, Dengue, MNV and YF] (Fig. 2b, d). This is consistent with the notion that certain defined structural determinants underlie the enhancement of pegRNA by 3' xrRNA-joining (discussions added in line 387-392).

Given our later mechanist data on the Zika motif, it is possible that different structural stabilities associated with these various motifs may shape their enhancement effect on pegRNA function. These comments are now added to the Discussion (line 435-438).

2. Line 109-110, correlates well might be an over statement since Zika shows higher in flow but less in Western.

[Authors' response]

We thank the reviewer for pointing out this problem. The flow data was presented as the relative percentage of EGFP⁺ cells, while the Western measures overall EGFP expression. The overall similarity in levels of MVE, WNV and Zika groups (see also NGS results in Fig. 1f) may have also brought some ambiguity to their cross comparisons. The phrase “correlates well” has now been changed into “roughly correlates” (line 109-110).

3. The authors may want to explain better why they chose to follow up with Zika xrRNA since the plots in Fig. 2 don't make this obvious.

[Authors' response]

We thank the reviewer for this question. From our initial characterization of editing effects at 6 sites, the Zika element showed the best overall enhancement effect in the contexts of PE2 and PE3. This comment is added to the Results (line 167-170). We have also pointed out the closely following MVE and Dengue groups in the context of PE3 results (line 160-163).

Although the functional differences among various xrRNA groups in Fig. 2 appeared modest, it is interesting that the results from both PE2 and PE3 experiments showed similar patterns of relative performances by different viral motifs [both featuring a descending order of Zika, MVE, Dengue, MNV and YF] (Fig. 2b, d). Such corroborative patterns from independent comparisons also strongly support the hitherto best-performing Zika xrRNA motif as an effective pegRNA potentiator, which we subsequently adopted for constructing xr-pegRNAs throughout the study. Related comments are added in the Discussion (line 387-395).

- 1 Nelson, J. W. *et al.* Engineered pegRNAs improve prime editing efficiency. *Nature Biotechnology*, doi:10.1038/s41587-021-01039-7 (2021).
- 2 Petri, K. *et al.* CRISPR prime editing with ribonucleoprotein complexes in zebrafish and primary human cells. *Nature Biotechnology* **40**, 189-193, doi:10.1038/s41587-021-00901-y (2022).

Reviewers' Comments:

Reviewer #1:

Remarks to the Author:

The author has fully addressed my concerns.

Reviewer #2:

Remarks to the Author:

The authors have responded in a satisfactory way to the reviews. The authors show that xrPE2 is similar in efficiency to the tools published by Nelson et al, so this paper will not provide any novel, improved tools. However, it will provide corroboration of this finding.